# Plasma Virome Reveals Blooms and Transmission of Anellovirus in Intravenous Drug Users with HIV-1, HCV, and/or HBV Infections

Yanpeng Li,[a] Le Cao,[a] Mei Ye,[b,c] Rong Xu,[a*] Xin Chen,[b,c] Yingying Ma,[a] Ren-Rong Tian,[b] Feng-Liang Liu,[b] Peng Zhang,[a] Yi-Qun Kuang,[d] Yong-Tang Zheng,[b,c] Chiyu Zhang[a]

[a]Shanghai Clinical Research Center for Infectious Disease (HIV/AIDS), Shanghai Public Health Clinical Center, Fudan University, Shanghai, China

[b]Key Laboratory of Animal Models and Human Disease Mechanisms of the Chinese Academy of Sciences, KIZ-CUHK Joint Laboratory of Bioresources and Molecular Research in Common Diseases, Center for Biosafety Mega-Science, Kunming Institute of Zoology, Chinese Academy of Sciences, Kunming, China

[c]University of Chinese Academy of Sciences, Beijing, China

[d]NHC Key Laboratory of Drug Addiction Medicine, First Affiliated Hospital of Kunming Medical University, Kunming Medical University, Kunming, China

Yanpeng Li, Le Cao, and Mei Ye contributed equally to this article. Author order is based on their contributions to the writing of the manuscript.

**ABSTRACT** Intravenous drug users (IDUs) are a high-risk group for HIV-1, hepatitis C virus (HCV), and hepatitis B virus (HBV) infections, which are the leading causes of death in IDUs. However, the plasma virome of IDUs and how it is influenced by above viral infections remain unclear. Using viral metagenomics, we determined the plasma virome of IDUs and its association with HIV-1, HCV, and/or HBV infections. Compared with healthy individuals, IDUs especially those with major viral infections had higher viral abundance and diversity. *Anelloviridae* dominated plasma virome. Coinfections of multiple anelloviruses were common, and anelloviruses from the same genus tended to coexist together. In this study, 4,487 anellovirus ORF1 sequences were identified, including 1,620 (36.1%) with less than 69% identity to any known sequences, which tripled the current number. Compared with healthy controls (HC), more anellovirus sequences were observed in neg-IDUs, and HIV-1, HCV, and/or HBV infections further expanded the sequence number in IDUs, which was characterized by the emergence of novel divergent taxons and blooms of resident anelloviruses. Pegivirus was mainly identified in infected IDUs. Five main pegivirus transmission clusters (TCs) were identified by phylogenetic analysis, suggesting a transmission link. Similar anellovirus profiles were observed in IDUs within the same TC, suggesting transmission of anellome among IDUs. Our data suggested that IDUs suffered higher plasma viral burden especially anelloviruses, which was associated with HIV-1, HCV, and/or HBV infections. Blooms in abundance and unprecedented diversity of anellovirus highlighted active evolution and replication of this virus in blood circulation, and an uncharacterized role it may engage with the host.

**IMPORTANCE** Virome is associated with immune status and determines or influences disease progression through both pathogenic and resident viruses. Increased viral burden in IDUs especially those with major viral infections indicated the suboptimal immune status and high infection risks of these population. Blooms in abundance and unprecedented diversity of anellovirus highlighted its active evolution and replication in the blood circulation, and sensitive response to other viral infections. In addition, transmission cluster analysis revealed the transmission link of pegivirus among IDUs, and the individuals with transmission links shared similar anellome profiles. In-depth monitoring of the plasma virome in high-risk populations is not only needed for surveillance for emerging viruses and transmission networks of major and neglected bloodborne viruses, but also important for a better understanding of commensal viruses and their role it may engage with immune system.

Address correspondence to Chiyu Zhang, zhangcy1999@hotmail.com, or Yong-Tang Zheng, zhengyt@mail.kiz.ac.cn.

*Present address: Rong Xu, State Key Laboratory of Proteomics, Beijing Proteome Research Center, National Center for Protein Sciences (Beijing), Beijing Institute of Lifeomics, Beijing, China.

The authors declare no conflict of interest.

**KEYWORDS** plasma virome, IDUs, viral expansion, anellovirus, transmission cluster, HIV-1, HCV

The virome is part of the microbiome, and it can be described as the collection of all the viruses, including prokaryotic viruses or bacteriophages, and eukaryotic viruses (1, 2). In the past decades, along with the development of viral metagenomics, substantial progress was achieved in the discovery of virosphere diversity and the uncovering of virus evolution (3–6), identification of viral etiologies responsible for specific symptoms or disease outbreaks (7–12), as well as the virome-host interactions and potential role it may engage with the physiology and disease of the host (13–18). In the context of humans, the unbiased viral metagenomics greatly improves our recognition of the viral sequences present in both healthy individuals and those with different disease status, and is increasingly accepted and used in clinical settings (19–21).

Previous studies of the human virome mainly focused on gut, respiratory tract, and skin (22–27). Blood virome consist of many commensal viruses, as well as pathogenic viruses (21). Certain primary viral infection leads to altered plasma virome. For example, an increased viral burden (e.g., anellovirus and human endogenous retrovirus or HERV) was associated with HIV-1 infection in certain populations (28, 29). The altered virome is associated with immune status and disease progression, which raises questions about whether resident plasma viruses could participate in immune response and influence health conditions (30). Among these resident plasma viruses, *anelloviridae* is the most abundant and widespread virus family. Anellovirus is a single-stranded circular DNA (ssDNA) virus, which was first reported in hepatitis patient. Subsequent studies revealed a commensal nature of anellovirus because the virus circulates in both health people and those with disease. Even though no association of anellovirus with specific disease was established, it was reported to engage with our host immune system, and could be used as surrogate marker for immune status (31, 32). Great diversities of anelloviruses have been reported in human blood. In particular, a recent study nearly tripled the whole anellovirus genomic sequence number. These data highlighted uncovered diversity and the mystery nature of these viruses (33).

Besides, characterizing the blood virome is of great importance for surveillance of pathogenic viruses and for transfusion safety. Unrecognized or neglected viral infections can be present without symptoms or with mild symptoms and spread to other individuals through blood transfusion or organ transplantation, which may lead to serious complications in the recipients (20, 34). Intravenous drug users (IDUs) are one of the major routes of acquiring main blood transmitted viral (HIV-1, HCV, and HBV) infections worldwide (35, 36) due to frequent needle sharing and unsafe syringe cleaning practices (37). Besides, as these population may live with risk behavior, poor health outcome or even suboptimal immune status, they may also tend to acquire other viral infections and even carry novel viruses.

The metagenomics era has unmasked unprecedented diversity of viruses, and makes it possible to study viruses in human health with a new ecological perspective (38–41). In this study, we used unbiased viral metagenomics to explore the plasma virome of individuals who are at high risk of potential exposure to viral infections through long-term injection drug use. We compared the virome of individuals who were previously identified to have HIV-1, HCV, and/or HBV infections to determine whether these major blood transmitted viruses would have any influence on their plasma viral composition and diversity, as well as revealing additional or novel viruses that may be neglected but are of potential risk to human health. Besides, by analyzing the diversity and evolution of the most abundant and widespread anellovirus in detail, we provided a comprehensive knowledge of their dynamics in the plasma virome of IDUs.

## RESULTS

**Participants.** Of 99 plasma samples from IDUs, 26, 11, and 10 were detected as HIV-1, HCV, and HBV positive, respectively, 29 were identified as positive for both HIV-1 and HCV,

**TABLE 1** Characteristics of healthy participants and IDUs in this study

| Sociodemographics | Healthy control[a] | Intravenous drug users (IDUs) | | | | |
|---|---|---|---|---|---|---|
| | | Neg-IDUs[d] | HIV-1+ | HCV+ | HIV-1/hCV+ | HBV+ |
| No. of individuals | 11 | 23 | 26 | 11 | 29 | 10 |
| Age | 31 (19 to 60) | 30 (21 to 37) | 31 (22 to 53) | 29 (21 to 39) | 31 (18 to 43) | 30.5 (28 to 36) |
| Gender (male/female) | 11/0 | 23/0 | 20/0 | 10/1 | 25/4 | 11/0 |
| Higher education | 9.1% | 0 | 0 | 0 | 0 | 0 |
| Married | 10 | 8 | 7 | 4 | 6 | 2 |
| Single/divorced | 1 | 15 | 20 | 7 | 23 | 8 |
| Duration of drug use/y | | 3.5 (1 to 16) | 8.0 (1 to 20) | 4.0 (1 to 13) | 10.0 (1 to 22) | 6.0 (2 to 15) |
| Frequency of drug use (times/day) | | 3.0 (0.3 to 7) | 3.0 (1.5 to 10) | 3.0 (1 to 3) | 3.5 (1 to 9) | 4.0 (1 to 5) |
| Syringe sharing | | 38.1% | 60.0% | 30.0% | 50.0% | 77.8% |
| Times of repeated syringe use | | 1.0 (1 to 10) | 2.0 (1 to 8) | 1.5 (1 to 5) | 2.0 (1 to 10) | 3.0 (1 to 5) |
| Cleaning method | | | | | | |
| Tap water | | 75% | 50% | 66.7% | 78.6% | 71.4% |
| Boiled water | | 25% | 28.6% | 16.7% | 7.1% | 14.3% |
| Never | | 0 | 21.4% | 16.7% | 14.3% | 14.3% |
| STD[b] history | | 10.0% | 5.9% | 33.3% | 0 | 22.2% |
| Sexual partners | | 3.0 (0 to 10) | 1.0 (0 to 6) | 2.0 (0 to 5) | 1.0 (0 to 5) | 3.0 (1 to 10) |
| Ways of drug use | | | | | | |
| Intravenous | | 33.3% | 23.8% | 70.0% | 65.0% | 44.4% |
| Intravenous and oral | | 66.7% | 76.2% | 30.0% | 35.0% | 55.6% |
| Drug rehabilitation times | | 2.0 (1 to 4) | 2.0 (1 to 15) | 2.0 (1 to 5) | 4.0 (2 to 15) | 2.5 (1 to 5) |
| Other drug use[c] | | 74.7% | 75.0% | 85.7% | 78.9% | 100% |

[a]Healthy control group (non-IDU and without main viral infections).
[b]STD, sexually transmitted disease.
[c]IDUs without HIV-1/HCV/HBV infections.
[d]Other drugs include Diazepam and Triazolam.

and other 23 were negative for all three viruses (neg-IDUs) by RT-qPCR assay (Table 1). The subjects had a median age around 30, and the vast majority (95%) of them are male. Only one person from HC had higher education experience, and all the IDUs had primary or middle school education experience. Ten out of 11 healthy individuals were married, while the majority (73%) of the IDUs were either single or divorced. Compared with neg-IDUs, IDUs positive for HIV-1, HCV, and/or HBV had longer history and higher frequency of drug use, higher rates of syringe sharing and repeated use, and poor cleaning and disinfection practice (Table 1).

**Overview of the plasma virome of IDUs.** Unbiased viral metagenomic method was used to determine the viromes of all plasma samples, including centrifugation, filtration, nuclease digestion, followed by random amplification, library preparation, and sequencing (see Materials and Methods). In total, 1.12 billion paired-end clean reads were obtained with an average of 10 million reads for each sample. After removing host and bacterial sequences, 29.9% of the reads were classified as viral sequences. In general, the plasma virome of the HC and IDUs were dominated by eukaryotic viruses (88.6% of total viral reads), followed by prokaryotic viruses (bacteriophages, 10.7%) and unclassified viruses (0.7%) (Fig. 1a). Vertebrate viral reads accounted for the vast majority (98%) of eukaryotic viral reads, and the rest of the reads were from plant, fungi and invertebrate host origins. We identified the presence of main vertebrate viral families, including *Anelloviridae*, *Flaviviridae*, *Retroviridae*, *Hepadnaviridae*, *Circoviridae*, *Orthomyxoviridae*, *Genomoviridae*, *Papillomaviridae*, *Pneumoviridae*, *Parvoviridae*, and *Kolmioviridae*. *Anelloviridae* and *Flaviviridae* were the most prevalent viruses, and they accounted for 84% and 15.9% of all vertebrate viral abundance, respectively (Fig. 1a).

We then compared the sensitivity of metagenomic sequencing with the qPCR results (Table S1). HIV-1 was detected in 24 out of 26 samples in HIV-1+ group, HCV in 11/11 for HCV+ group, HIV-1 and HCV in 28/29 for both HIV+ and HCV+ group, and HBV in 9/10 for HBV+ group. These data indicated that the metagenomic sequencing

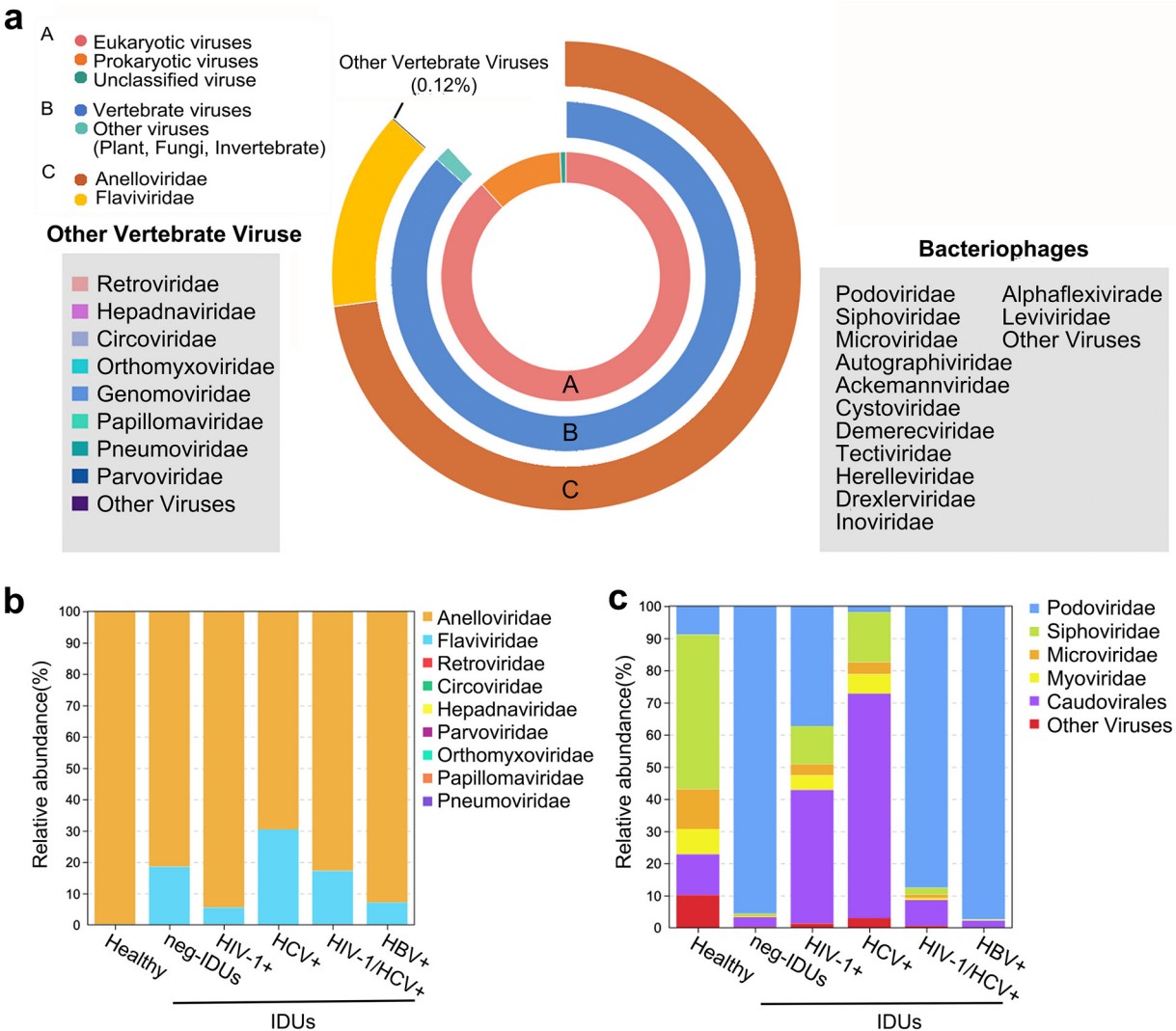

**FIG 1** Summary of the viral compositions detected in this study. The donut chart shows the distribution of all viral reads according to taxonomical ranks (a). Relative abundance of main vertebrate viruses (b) and prokaryotic viruses (bacteriophages) (c) in different groups.

method had a good sensitivity for the detection of different viruses. Expect for HBV, no significant correlations between the number of viral reads and Ct values were observed for HIV-1 and HCV (Fig. S1a).

Plasma viromes of both healthy individuals and IDUs were dominated by anellovirus (Fig. 1b; Fig. S2a), and a significant correlation between the number of viral reads and Ct values was observed for anellovirus (Fig. S1b). Compared with HC, plasma virome of IDUs, regardless of infection or free for HIV-1, HCV and/or HBV, were more likely to carry other viruses, such as pegivirus (35 in IDUs, one in HC), *parvoviridae* (20 in IDUs, 0 in HC), and *circoviridae* (34 in IDUs, 3 in HC) (see supplemental material). The distribution of plasma bacteriophages was relatively even in healthy individuals, but was characterized by one or two dominant bacteriophage taxons in IDUs (*Podoviridae* and *Caudovirale*) (Fig. 1c; Fig. S2b).

To investigate the plasma viral burden and complexity in HC and IDUs, we determined the viral abundance (reads per million, RPM) and viral diversity (richness, Simpson and Shannon indexes) of different groups. Compared with HC, the plasma of neg-IDUs had a relatively higher total viral abundance ($P = 0.20$), and IDUs with main viral infections appeared to have even higher plasma viral abundance (Fig. 2a). Besides, richness score showed a similar trend of the distribution of viral taxons. HC had the lowest numbers of plasma viral taxons, neg-IDUs, and those with main viral

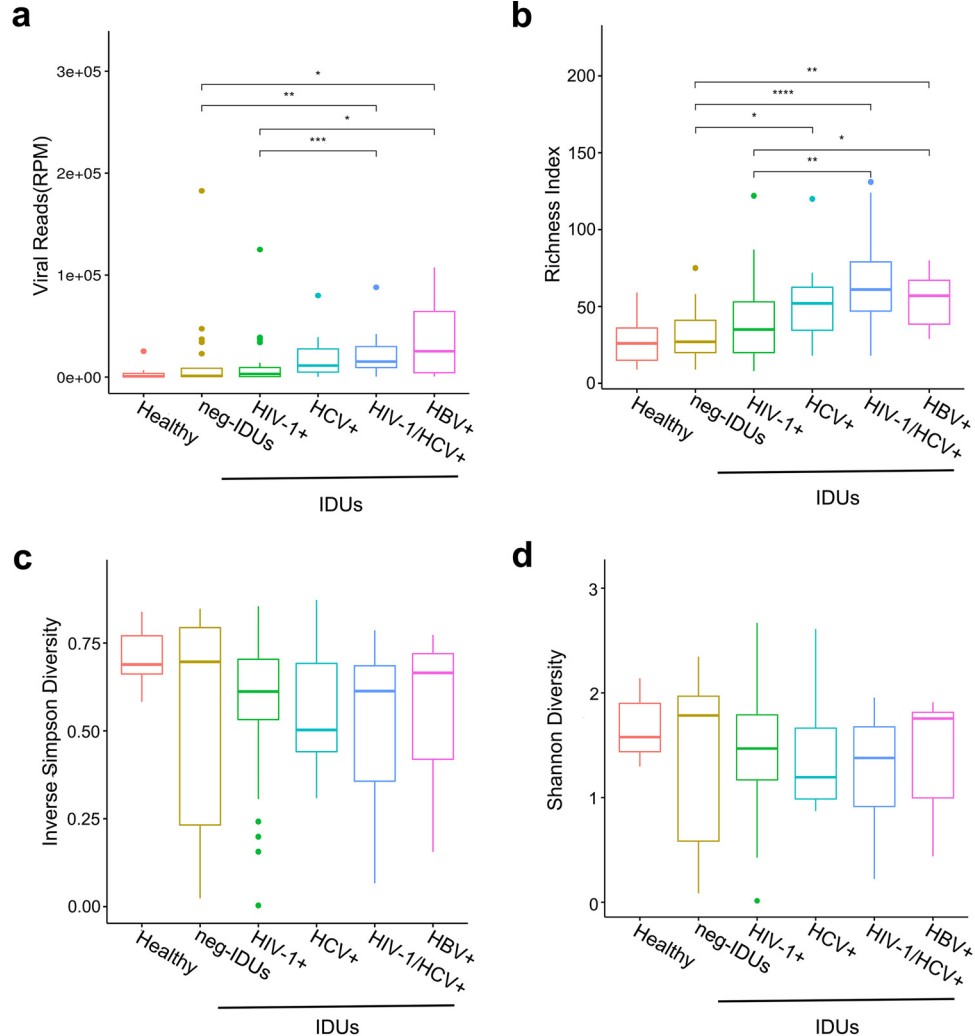

**FIG 2** Viral abundance and diversity among different groups. Viral abundance was shown as reads per million (RPM) (a). Viral diversity was shown for each group with richness score (number of annotated taxons) (b), Inverse Simpson index (c), and Shannon index (d).

infections showed increased numbers (Fig. 2b). As for both Inverse Simpson and Shannon indexes, there were no apparent differences between HC and neg-IDUs, but IDUs with viral infections showed lower scores (Fig. 2c and d), which indicated unevenly distributed viral populations possibly due to the expansion of certain viruses. In addition, we observed significant positive correlations between duration of drug use and both viral abundance and richness (Fig. S3). There was no influence of age on the composition of plasma virome.

**Profile of the plasma *Anelloviridae*.** As the anelloviruses were the most abundant and prevalent eukaryotic viruses in plasma virome, we further characterized their diversity and distribution among different groups. *Anelloviridae* was present in all the plasma samples tested, and at least two different anellovirus taxons could be annotated in each sample. Over 71% of the samples had more than 10 anellovirus taxons, and 8.9% had over 50 different anelloviruses. Compared with HC, IDUs had more anelloviruses in individual and group levels. In particular, one sample from IDUs was found to carry 65 different anelloviruses (Fig. S4a, b). These results indicated a great diversity and individual variation of plasma anelloviruses. The number of anelloviruses identified among neg-IDUs did not significantly differ from that in HC, while infection with HIV-1, HCV, and/or HBV significantly increased the number of anelloviruses in IDUs (Fig. 3a). For the three main genera, *alphatorquevirus* was the most abundant genera among all

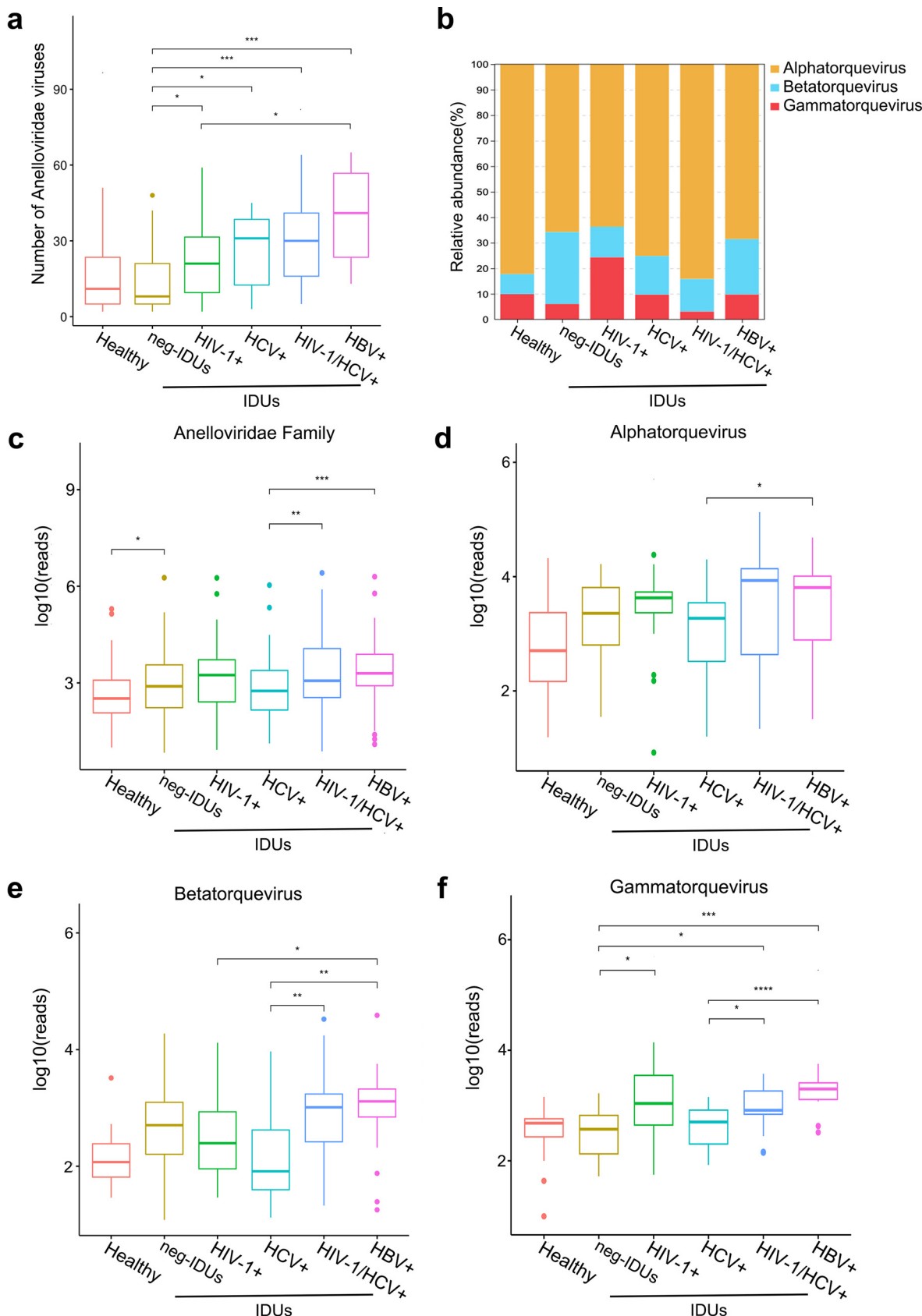

**FIG 3** Expansion and distribution of anelloviruses among different groups. Number of annotated anelloviruses (a). Relative abundance of three anellovirus genera in different groups (b). Abundance ($\log_{10}$ reads) of *Anelloviridae* (c), *alpha*-(d), *beta*-(e), and *gammatorquevirus* (f).

samples. Compared with HC, a slight increase of relative abundance of *betatorquevirus* was observed in neg-IDUs ($P > 0.05$) (Fig. 3b), and similar increase were also observed for the absolute reads number of total anellovirus, *alphatorquevirus*, and *betatorquevirus* genera (Fig. 3c to e). Expect IDUs with HCV infections, higher abundance of not only total anellovirus, but also *alphatorquevirus* and *gammatorquevirus* genera was associated with HIV-1 and HBV infections, as well as HIV-1/HCV coinfections (Fig. 3c, d, and f). For *betatorquevirus* genera, its increase was only observed in IDUs with HIV-1 or HBV infections (Fig. 3e).

**The enormous genetic diversity and expansion of *Anelloviridae*.** Due to the extremely high genetic diversity of anelloviruses, different subtypes/strains within the same species are common. In order to reflect the actual diversity and potential novel species of the *anelloviridae*, we extracted all the ORF1 sequences and analyzed their distribution and diversity among different groups. In total, we obtained 4,487 ORF1 sequences (over 1,500 bp) and found that ORF1 sequences heavily outnumbered annotated anellovirus taxons (Fig. 3a and 4a), which indicates a significant variability of anellovirus genomes. Compared with healthy individuals, neg-IDUs showed a higher number of ORF1 sequences, and the numbers were even higher in IDUs with HIV-1, HCV, and/or HBV infections, which are very similar to the observation in the distribution of anellovirus abundance (Fig. 3c). We then performed a phylogenic analysis with maximum-likelihood method using all the ORF1 sequences together with reference sequences from the three main genera, and found that all these sequences could cluster into the three main genera (Fig. 4b). Most sequences clustered closely with references; but the other sequences either did not cluster with known references, or formed independent phylogenetic clade, suggesting high genetic diversity of anellovirus and the presence of potential new species. Alignment analysis showed that 63.9% of the ORF1 had identities over 69% to all known anellovirus sequences in the database, and the rest ($n = 1,620$) were potential novel species with identities below 69% (Fig. 4c). Further analyses of all these ORF1 sequences showed an expansion of both novel and known anelloviruses in plasma of IDUs, especially those with HIV-1, HCV, and/or HBV infections (Fig. 4c).

Next, we measured the pairwise distance of ORF1 sequences within each individual to determine genetic variation. Even though in HC whose anellovirus abundance is relatively low, the pairwise distance remained high (mean distance of 0.645). Neg-IDUs showed a slightly higher (0.658) pairwise distance than HC. Consistent with the abundance and number of anellovirus sequences, IDUs with HIV-1, HCV, and/or HBV infections had the highest plasma virus pairwise distance (mean distance from 0.673 to 0.688) (Fig. 5a). These data indicated the expansion and diversification of anelloviruses after certain viral infections in IDUs.

To determine whether the virus expansion favors certain viruses and how different viruses coexist with each other, we further analyzed ORF1 sequence clusters according to their genetic identities. Healthy individuals had more sequences that fell into clusters with relatively low identities (<75%), while IDUs with or without main viral infections had more sequences belonging to higher identity clusters (≥85%), and all sequences in the clusters with ≥95% identity were found in IDUs (Fig. 5b). This result revealed that HIV-1, HCV, and/or HBV infections in IDUs may lead to the expansion of anellovirus and form many clusters with high genetic identities.

Even though most of the anelloviruses were shared by different individuals (more than 71%) (Fig. 5c), we found 12 anelloviruses were only detected in IDUs with major viral infections, and no anelloviruses were found to be specifically associated with certain viral infections (Fig. 5d). The 12 anelloviruses were mainly from *alphatorquevirus* and *betatorquevirus*, and different from the anelloviruses that fell into the >95% identity clusters (Fig. S4c; Table S2). These data further indicated the expansion of anellovirus after viral infection may lead to both the emergence of new anelloviruses and blooming of existing anelloviruses. Using co-occurrence network analysis, we found that the same anellovirus genera tended to form close networks and coexist together (Fig. 5e). The main hubs of the *alphatorquevirus* (purple) and *gammatorquevirus* (green)

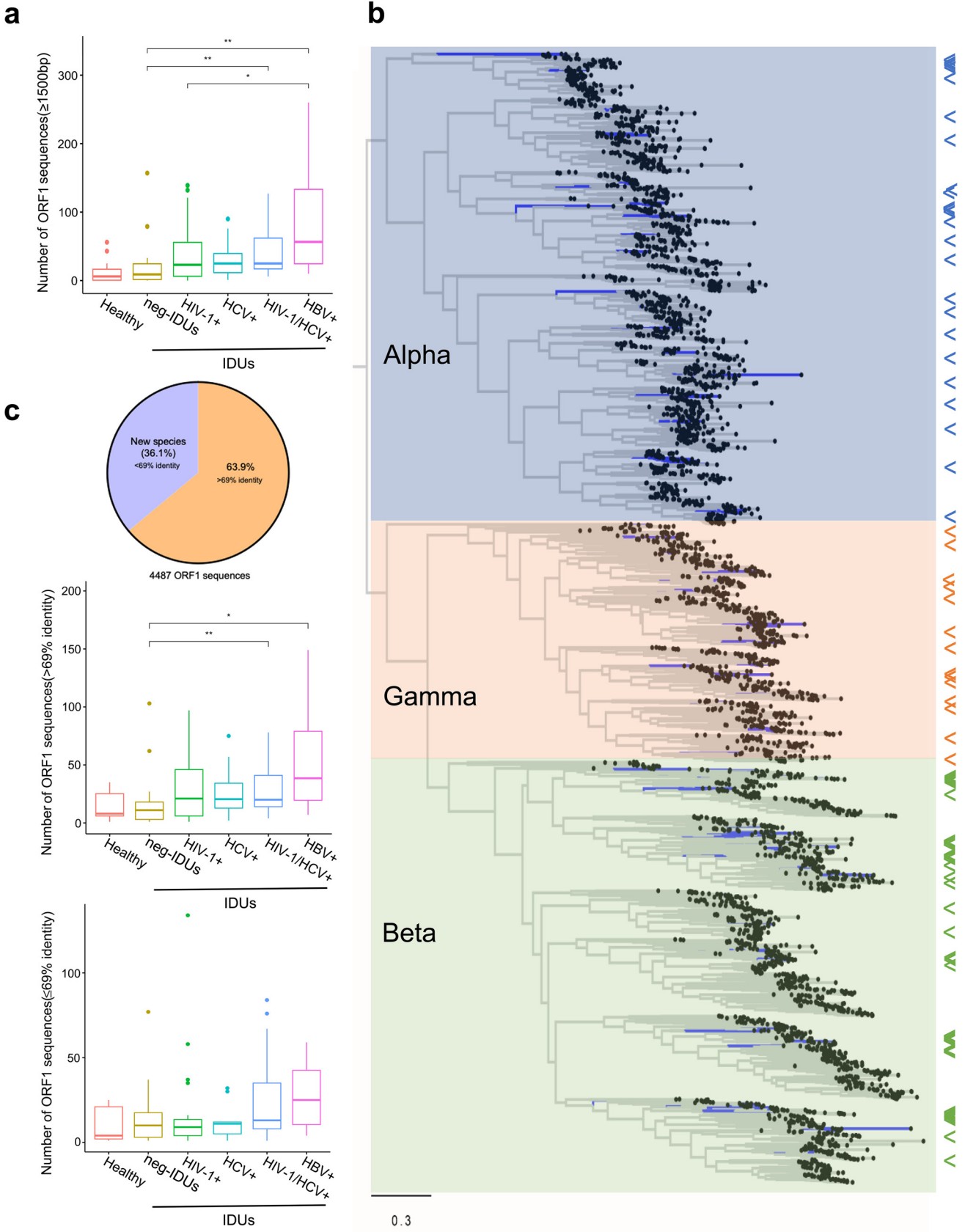

**FIG 4** Unprecedented plasma anellovirus diversity. Number of anellovirus ORF1 sequences ($>$1,500 bp) in each group (a). Maximum-likelihood phylogenetic tree of ORF1 sequences. Arrows to the right of the tree indicate the positions of reference sequences (b). Distributions of new ORF1 sequences with over or below 69% identity to all currently known anelloviruses (c).

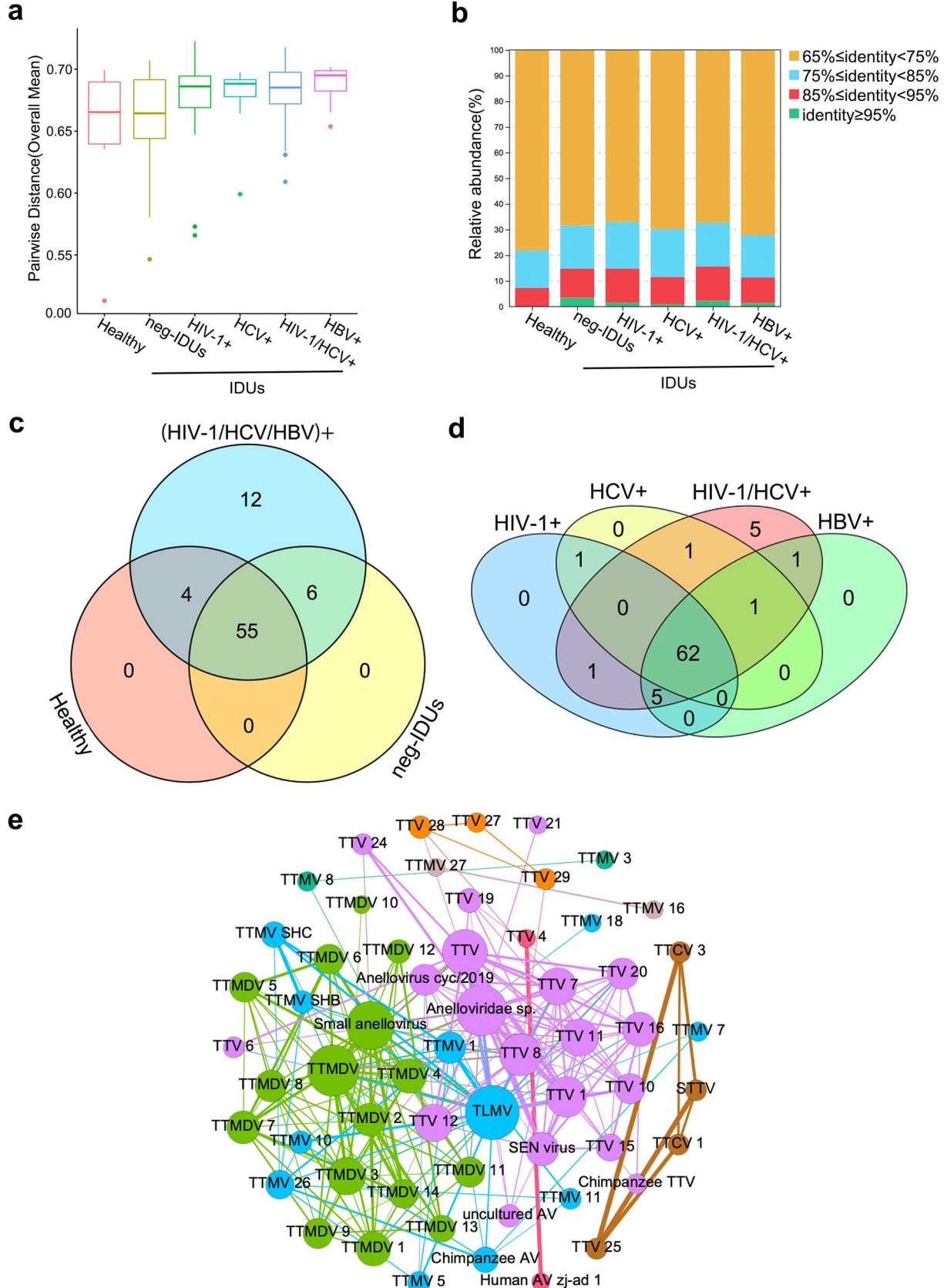

**FIG 5** Blooms and coexistence of certain anelloviruses. Comparison of within individual pairwise distances between different groups (a). Distributions of ORF1 sequence clusters that have different identities (b). Venn diagrams of different anellovirus taxons between healthy

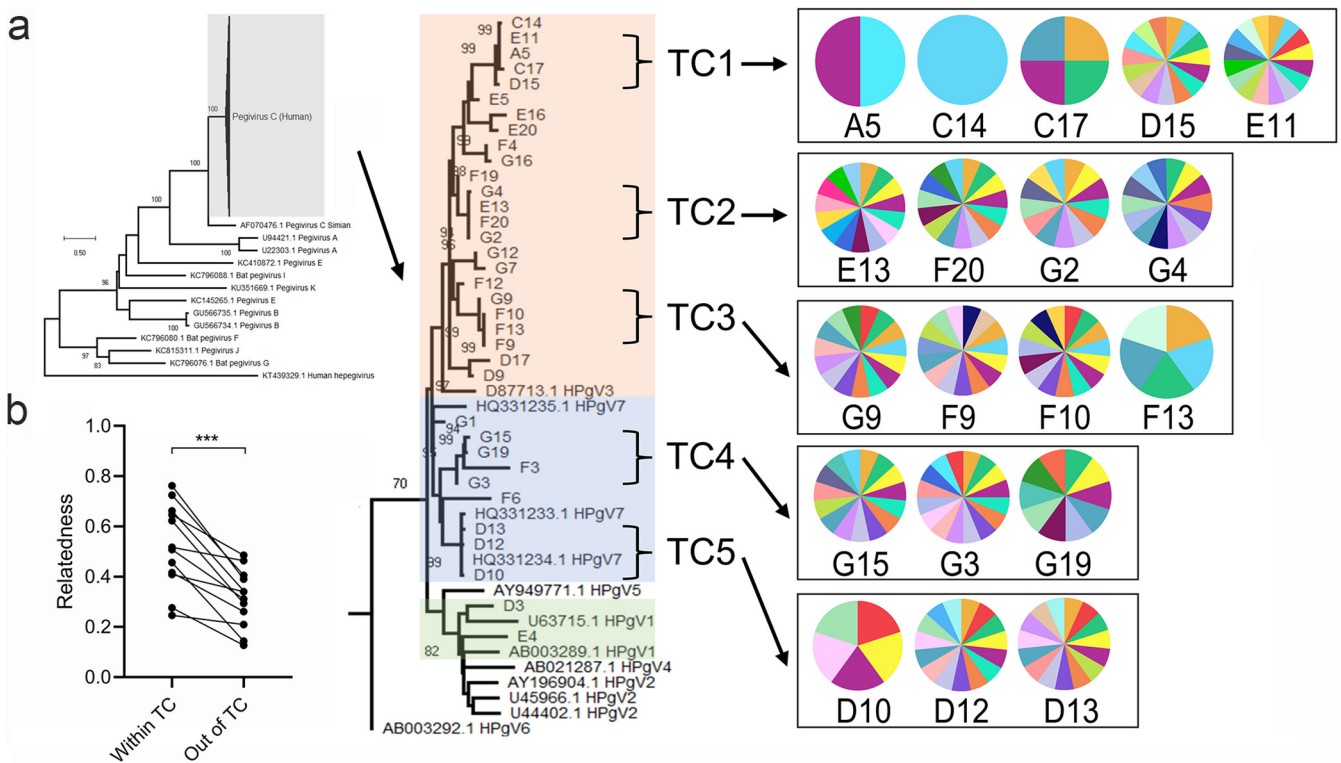

**FIG 6** Transmission of pegivirus and anellovirus among IDUs. (a) Maximum-likelihood phylogentic tree of human pegivirus C. Three human pegivirus genotypes are highlighted with orange, blue, and green colors. Five transmission clusters (TC1 to 5) are labeled. Pie charts to the right show the individual anellome profile within each TC (presence of each anellovirus), only top 15 abundant anelloviruses are shown. (b) Relatedness of anellovirus reads within TC versus unrelated individuals (also described in Table S3). Relatedness was compared with Wilcoxon matched-pairs signed rank test.

networks included *Torque teno virus* (TTV) 1, 7, 8, and 11 and *Torque teno midi virus* (TTMV) 2, 3, and 4, as well as two unclassified viruses in each network, respectively. *TTV-like mini virus* (TLMV) dominated the *betatorquevirus* (blue) networks. Eight of the 12 anelloviruses only detected in IDUs with major viral infections formed several independent clusters (Fig. S4d).

**Transmission of pegivirus and anallovirus in IDUs.** Pegivirus belongs to *Flaviridae*. Human pegivirus (HPgV) has high prevalence in HIV-1 and HCV infected individual, and shows a beneficial effect on HIV-1 infection (42). In this study, HPgV was detected in 37 individuals, and most of these individuals were IDUs with HIV-1, HCV and/or HBV infections (Fig. S5). Of all the pegivirus, 35 were HPgV-C, and two were HPgV-H (or HPgV-2). Only one healthy individual and five neg-IDUs were also positive for pegivirus. The detection rate of HPgV was well consistent with its prevalence in these populations (42). We previously reported significantly negative correlations of HPgV abundance with HIV-1 and anellovirus abundance in men who have sex with men (MSM) (30), while the correlations were not found in IDUs ($P = 0.53$ and $0.76$) (Fig. S5). Only a weak positive correlation between pegivirus and HCV ($P = 0.27$) was detected (Fig. S5). Phylogenetic analysis of full-length NS5B sequences of human HPgV-C showed that most of them belonged to HPgV3/7 genotypes (Fig. 6a), which was consistent with the prevalence of both pegivirus genotypes in this region (43). HPgV-C genotype 1, which was rarely detected in China, was found in two individuals.

**FIG 5** Legend (Continued)

individuals, neg-IDUs, IDUs with HIV-1/HCV/HBV infections (c), and IDUs with different infection patterns (d). Co-occurrence network of different anelloviruses. Viruses in the same network were highlighted with same color. Bigger size of the circle means more viruses it interacts with, and the line between two circles indicates the frequency of the co-occurrence. TTV, Torque teno virus; TTMV, Torque teno mini virus; TTMDV, Torque teno midi virus; TLMV, TTV-like mini virus; STTV, Simian torque teno virus; AV, anellovirus.

In order to determine whether HPgV identified among these IDUs had transmission links, we performed transmission cluster analysis based on genetic distances of closely related sequences. At least five transmission clusters that contained 3–5 HPgV variants with more than 99% identity were identified. In total, 54.3% ($n = 19$) of HPgV sequences formed TCs, and 18 of them were from IDUs. These data indicated that HPgV, like HIV and HCV, was mainly transmitted among IDUs in transmission cluster manner (Fig. 6a). In particular, TC5 contained two references that were isolated from IDUs in the same region of Yunnan, supporting the local transmission link of HPgV among IDUs.

As the transmission of anelloviruses was common in blood transfusion or organ transplantation (44), we asked whether anellome is transmissible among IDUs. Because the identification of TCs of HPgV allowed us to determine the IDUs who have potential transmission and epidemiological links, we further characterized and compared the anellovirus profile among each individual within the five TCs. Two to three IDUs in each TC shared highly similar anellovirus profiles that shared 67% to 99% of annotated anelloviruses. For example, two individuals in TCs 1, 4, and 5, and three IDUs in TCs 2 and 3 displayed highly similar anellovirus profiles (Fig. 6). Of particular importance is that two pairs of IDUs in TC3 shared 100% the same anelloviruses (Table S3). One healthy individual in TC1, and one neg-IDUs in TC1 and TC3 displayed relatively "simple" anellome, which was consistent with the relatively low diversity of anellovirus in general population and neg-IDUs (Fig. 2b and 3a). However, different individuals showed great variations in anellovirus abundance, even between those had similar anellome profiles (Fig. S6). As more than 36% of anelloviruses found in this study were potential novel species, and in order to reflect the transmission of anelloviruses more accurately among IDUs, we compared the relatedness of all anelloviruses (both annotated and undefined) within or outside the TCs defined by pegivirus. Compared with unrelated individuals outside each TC, individuals from the same TC that shared similar anellome profiles had more related anelloviruses ($P = 0.0005$) (Table S3; Fig. 6b). Both results from Fig. 6 and Table S3 indicated the transmission of anelloviruses along TCs in IDUs.

## DISCUSSION

In this study, using viral metagenomics we explored the plasma virome of IDUs, and whether the changes of viral compositions circulating in these individuals was associated with HIV-1, HCV, and/or HBV infections. IDUs generally bear a high burden of the main bloodborne viruses (e.g. HIV-1/HCV/HBV), and injection drug use as well as other risk practices are major drivers of new infections and transmissions globally (45–48). Relatively low viral population diversity, high viral taxon number and viral abundance in IDUs, especially those with major viral infections, indicated higher viral burden of different viruses and probably the expansion of certain viruses in the blood of IDUs. We observed higher proportion of risk behaviors in IDUs with HIV-1, HCV, and/or HBV infections than healthy individuals, including longer duration of drug use, higher frequency of syringe sharing and repeated use, and worse syringe cleaning practices. In particular, longer duration of drug use was found to be associated with higher viral abundance and richness, indicating that long-time drug abuse would change the virome composition probably due to higher risk of exposure to various viral infections. Besides, all the IDUs were from the underdeveloped regions of Yunnan, near the China-Myanmar border, which is close to the "Golden Triangle," where active trading, travelling, and syringe sharing between different risk populations would be bridges for further transmissions (49, 50).

*Anelloviridae* is the most ubiquitous and abundant viral family, and infections by anelloviruses are most probably asymptomatic (32, 51). Even though anelloviruses are generally known as a part of the commensal virome, numerous studies found the association of anellovirus with host immune status, and suggest anellovirus to be considered as a potential marker for immunocompetence in clinical organ transplantation (52–54). Anellovirus has also been associated with various liver diseases, such as acute

or transfusion associated hepatitis, and chronic hepatitis B and C (55). Compared with healthy individuals, the increase of anellovirus abundance in IDUs might indicate a poor health condition, and therefore a higher risk of infection by various viruses. The infections with HIV-1, HCV and/or HBV may have impaired the immune system, and make the latter unable to control the replication of some resident viruses such as anelloviruses. This is particularly true as some previous studies have consistently found that increased anellovirus load was associated with HIV-1 infections, and the anellovirus load was inversely correlated with CD4$^+$ T cell counts in MSM (30, 56).

Anellovirus is one of the most divergent viruses ever discovered, and has higher genetic diversity than HIV and papillomavirus. A recent study focusing on blood transfusion cohorts reported more than 1,600 anellovirus sequences, which nearly tripled the number of previously known anellovirus sequences (33). In this study, we obtained more than 4,450 ORF1 sequences, which further nearly tripled current anellome database. More than 36% of them have below 69% identity to known anelloviruses, unveiling the vast diversity of anellovirus. The results indicated that an expanded diversity and pairwise distance, as well as an increased abundance of anellovirus, was associated with injection drug use and major viral infections. Whether the blooms of anellovirus diversity and population size during HIV-1/HCV/HBV infections are a consequence of fast mutation rate and/or frequent recombination between different variants need further analyses (33, 57).

Because IDUs are the main population for transmission of many blood-borne viruses, we asked whether the anellome could be transmitted like other viruses. Even though viral abundance varied largely in different individuals, the similar anellovirus profiles in IDUs within the HPgV transmission clusters suggested that anelloviruses can be transmitted among these IDUs like other common virus (e.g., HIV-1, HPgV). However, not all the individuals within each transmission cluster showed the same anellome profile. There may have two explanations. First, the syringe sharing could happen before HPgV infections, and the "donor" may have a different anellome. Second, the anellome was similar between each individual at the beginning after transmission, but after a long period interaction with host's immune system, it evolved to a different profile, as different individuals may favor certain virus types. Furthermore, these data also suggested a potential use of commensal viruses to trace transmission networks of main blood-borne viruses. For example, a recent study suggested that anellovirus could help to reveal drug use network, and early accumulation of anellovirus may predict the infection risk of HIV-1 and HCV (58).

We further confirmed the huge and yet to be discovered diversity of anelloviruses. The expansion and transmission of anelloviruses in these IDUs after main viral infections showed the ubiquity and active replication of these viruses within the host. We didn't know whether the blooms of anellovirus would have any influence on either other viruses' replication or disease progression, but their unique traits could provide opportunities for potential translational applications. For example, the sensitivity of anellovirus's response to immune status or possible disease progression implied its diagnostic potential to support the development of personalized treatment strategies (51, 52). Besides, commensalism, transmission, and persistence of diverse anelloviruses imply less or no damage to the host, making them attractive vector candidates for delivering therapeutic strategies for different diseases. We could imagine more anelloviruses will be unveiled, and whether different species or genotypes of anelloviruses have different tissue or cell tropisms, and then influence host immunity and health condition need further studies.

HIV-1/HCV/HBV are globally screened, and the drugs and prevention measures against these viruses are available. However, many emerging and neglected viruses could be circulating in IDUs. These viruses may have potential disease outcomes or pose threats to others through blood transmission (55). In this study, several other blood-borne viruses that may associate with hepatitis were identified, including human pegivirus C (HPgV-1), more recently discovered pegivirus H (HPgV-2), and hepatitis

delta virus (HDV) (Table S4). Even though no pathogenicity of HPgV is established, growing lines of evidences confirmed its high prevalence in persons with blood-borne or sexually transmitted infections. However, a lot of previous studies indicated a beneficial effect of HPgV during HIV-1 infection (42, 59). Whether the coinfections of HPgV with other viruses would impact the health outcome needs further investigation. HDV depends on HBV for propagation and may lead to the most severe form of viral hepatitis (60). The detection of HDV together with HPgV, HIV-1, and HCV in IDUs showed a complex circulation of blood-borne viruses and raised concerns about severe disease outcome and transmission risk among these populations. Besides, we were able to assemble a full circular viral genome (GenBank: ON226770) in one IDU sample, and this genome was most closely related to porcine circovirus 3 with only 70% identity, implying that it may be a novel circovirus specific to humans. The presence of these potential emerging viruses in this high-risk group highlighted the importance to screen and investigate their prevalence in IDUs and other high-risk cohorts, and to further determine their origin and transmission pattern (61).

There were several limitations of this study. First, the number of individuals in several groups was relatively small. Second, compared with many virome studies that focused on either RNA or DNA viruses, we employed an unbiased method targeting both RNA and DNA viruses. The strategy is time-saving, but the random amplification method used here may lead to further enrichment of circular genome sequences of *Anelloviridae*, which may disturb the relative viral abundance of some viruses. A methodology comparison between the methods with and without random amplification in performance will be very helpful in future study. Furthermore, even though the method used here was generally sensitive, as the virome composition may vary due to different populations, it is important to repeat a similar study and validate the findings in another cohort. Third, the lack of longitudinal samples limits us for further analysis of the virome dynamics within each individual.

**Conclusions.** Using viral metagenomics, this study unveiled the vast diversity of plasma virome in IDUs, and the blooms of anelloviruses may be associated with certain viral infections. Complex expansion and transmission of anelloviruses raised concerns about whether they may influence the susceptibility of IDUs and other groups to the infection by other viruses and disease outcome. In-depth monitoring of plasma virome in high-risk populations might not only be needed for both surveillance for emerging viruses and transmission networks of major and neglected blood-borne viruses, but also be important for a better understanding of the commensal viruses and their role it may engage with our immune system.

## MATERIALS AND METHODS

IDUs were from a previous transmitting viral infections surveillance program (2009 to 2012) in Yunnan Province, China. This study was approved by the Ethics Committees of Kunming Institute of Zoology, Chinese Academy of Sciences, and complied with all relevant ethical regulations (SWYX-2008010, SMKX-20180102-178). Blood samples from healthy non-IDUs of the same community were collected as controls (HC). Oral or written informed consents were obtained from all participants before sample collection. At least 2 mL of blood was collected from each participant (99 drug users and 11 healthy controls), and plasma was separated within 24 h upon sampling. All plasma samples were stored at −80°C until use. Initial detection of HIV-1/HCV/HBV was done by reverse transcription-real-time PCR (RT-qPCR; Sansure Biotech, China).

**Sample processing, library construction, and sequencing.** Enrichment of encapsidated DNA and RNA viruses was performed as previously described (18, 62). Briefly, 200 $\mu$L plasma was thawed on ice and homogenized for 3 to 5min, and then the suspension was centrifuged at 12,000 $\times$ $g$ at room temperature for 15 min. The supernatant was passed through a 0.45 $\mu$m sterile filter to reduce background materials (Costar Spin-X centrifuge tube filters, Corning, USA). Filtrates were incubated with a cocktail of nucleases, including 15U Turbo DNase (Invitrogen, USA), 20U Benzonase (Novagen, Germany), and 20U RNase I (Promega, USA) for 2 h at 37°C. The reaction was terminated with 30 mM EDTA at 65°C for 10 min. Total nucleic acids (including both DNA and RNA) were extracted using QIAamp MinElute virus kit (Qiagen, Germany), which were then amplified using a random-amplification approach (REPLI-g Single Cell WTA kit, Qiagen, Germany) according to the manufactures' instructions. The amplified products were purified by QIAquick PCR purification kit (Qiagen, Germany). DNA libraries were prepared using NEBNext UltraII FS DNA library Prep Kit (Illumina, USA), quantified by Qubit3.0 (Invitrogen, USA) and sequenced on the Illumina Novaseq platform (Illumina, USA) with 2 $\times$ 150-bp paired reads.

**Virome bioinformatic analyses.** Sequencing data were analyzed using an in-house pipeline. The NGS raw data were filtered by Cutadapt v.1.18 (63) and Trimmomatic v.0.38 (64) by removing Illumina sequencing adaptor and low-quality sequences. Human- and bacterium-derived sequences were subtracted from the data by Bowtie2 v.2.3.4.3 (65). The remaining high-quality reads were *de novo* assembled by Megahit v.1.1.3 (66). Assembled contigs, as well as singlets, were mapped against the viral nucleic acid and protein database using BLASTn (E $< 10^{-10}$) and BLASTx (E $< 10^{-5}$) (DIAMOND v.0.9.24) (67), respectively. All the viral hit candidates were then searched against the NCBI nonredundant nt and nr database to remove reads or contigs that have higher similarity to sequences related to host, bacteria, fungi, plasmids, vectors, and other nonviral sequences than to viral sequences (false positives). To reduce the risk of potential cross-library contamination due to index-hoping (68), viruses with a read count less than 0.1% of the highest count for that virus among the other libraries was removed for subsequent analyses. Viral abundance was calculated by reads per million (RPM).

**Phylogenetic analyses.** The viral sequences of anellovirus and pegivirus were extracted from all viral contigs/reads, and were aligned to reference viral genomes to generate full or partial genomes using Geneious R11 program (69). ORF1 region of anellovirus was extracted using NCBI's ORF Finder tool using the "any sense codon" option, which were then curated by aligning against anellovirus reference sequences. All the obtained ORF1 sequences of anellovirus were deduplicated with a threshold of 99% identity. ORF1 region of the reference anellovirus and NS5B region of the reference pegivirus were downloaded from the NCBI's GenBank database. Viral nucleic acid sequences were first translated into amino acids and aligned using MAFFT (70). Phylogenetic trees were inferenced using the maximum likelihood method with IQ-Tree (71). Model test program was used to determine the best substitution model. Phylogenetic trees based on nucleotide sequences were generated using the bootstrap method (1,000 times) under a GTR+G model. According to previous studies of HIV-1 and HCV (72), we used a stringent threshold of 1% genetic distance for the detection of transmission clusters (TC) of pegiviruses.

**Anellovirus diversity and distance.** Chao Richness Score was used to compute the number of anellovirus species in each group. The diversity of anellovirus community were indicated by both the Shannon Diversity Index and Inverse Simpson Index. Pairwise distances of ORF1 amino acid sequences within each group were calculated by MEGA 7.0 (73). All the ORF1 amino acid sequences were clustered into different groups based on their identities to each reference (65% to 75%, 75% to 85%, 85% to 95%, ≥95%) using CD-HIT (74), and only clusters that had at least 10 sequences were used for downstream analyses. In order to determine the potential transmission of *Anelloviridae* among IDUs, the general diversity and relatedness of *Anelloviridae* between individuals within or out of the TC (determined by pegivirus phylogeny) were compared (44, 58). Anellovirus reads were mapped to corresponding anellovirus contigs using highly stringent threshold with Bowtie 2 (44), and relatedness was shown as the proportion of recovered anellovirus reads within/out of TC relative to cognate anellovirus reads (anellovirus reads that were recovered from contigs of each individual).

**Detection of anellovirus with qPCR.** Primers (forward: 5'-ACWKMCGAATGGCTGAGTTT-3', reverse: 5'-CCCKWGCCCGARTTGCCCCT-3') targeting the conserved UTR region of anelloviridae was used (52). A SYBR qPCR method was used for the detection. The reaction contained 10 $\mu$L of 2× SYBR Green Pro *Taq* HS Premix (AGBio, China), 1 $\mu$L of both primers (10 uM each), 3 $\mu$L template, and 5 $\mu$L H$_2$O. qPCR programs were as follows: 95℃ for 1 min and 30 s, 40 cycles at 95℃ for 15 s, and 63℃ for 1 min.

**Statistics.** Differences between groups were determined by the nonparametric Kruskal-Wallis Test, correcting for multiple comparisons with Dunn's procedure. A difference with $P < 0.05$ was considered to be statistically significant. Co-occurrence networks of different anelloviruses were illustrated by Gephi v.0.9.2. Relatedness of anellovirus reads within TC versus unrelated individuals was compared with Wilcoxon matched-pairs signed rank test. Spearman's correlation was calculated to determine the correlation between the qPCR and metagenomic sequencing results (RPM). Influence of age and time of drug use on the blood virus composition was also determined by spearman's correlations (age/time of drug use versus RPM, Richness, Shannon index). All statistical analyses were performed using RStudio v.3.8.

**Availability of data and materials.** All the raw sequencing data were deposited in the CNSA (https://db.cngb.org/cnsa/) of CNGBdb (project number CNP0002340) under the accession numbers CNS0468254-CNS0468363. Anellovirus and pegivirus sequences generated in this study can be found in the same project under the accession numbers CNS0483741-CNS0483743.

## SUPPLEMENTAL MATERIAL

Supplemental material is available online only.

**SUPPLEMENTAL FILE 1**, PDF file, 1.2 MB.

## ACKNOWLEDGMENTS

Thank for Kai Liu for the support and suggestions at the early stage of the project. Thank for Rui Yu for the support of the revision of the manuscript. We also thank Jian-Hua Wang at Guangzhou Institutes of Biomedicine and Health, Chinese Academy of Sciences, for his very helpful discussion and suggestions on the draft.

This study was approved by the Ethics Committees of Kunming Institute of Zoology, Chinese Academy of Sciences, and complied with all relevant ethical regulations (SWYX-2008010, SMKX-20180102-178).

The authors declare no conflict of interests.

This study was supported by National Natural Science Foundation of China (32170147, U1302224, 81271892), Shanghai Science & Technology Innovation Action Program (20MC1920100), Yunnan Key Research and Development Program (202103AQ100001, 202102AA310055), and the Key Program of Chinese Academy of Sciences (KJZD-SW-L11-02).

C.Z., Y.-T.Z., and Y.L. participated in study design; M.Y., X.C., R.-R.T., and F.-L.L. collected samples; R.X., Y.M., and P.Z. performed the main experiments; L.C. and Y.L. performed bioinformatic analyses; Y.L., L.C., and C.Z. interpreted the data; Y.L. and C.Z. wrote the manuscript; Y.-T.Z. and Y.-Q.K. contributed to data interpretation and manuscript revision; C.Z. and Y.-T.Z. supervised the study. All authors read and approved the manuscript.

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
