## [Reviewer comments · Microbiology Spectrum]

Microbiology Spectrum

Plasma Virome Reveals Blooms and Transmission of Anellovirus in Intravenous Drug Users with HIV-1, HCV and/or HBV infections

Yanpeng Li, Le Cao, Mei Ye, Rong Xu, Xin Chen, Yingying Ma, Ren-Rong Tian, Fengliang Liu, Peng Zhang, Yi-Qun Kuang, Yong-Tang Zheng, and Chiyu Zhang

Corresponding Author(s): Chiyu Zhang, Shanghai Public Health Clinical Center, Fudan University

Review Timeline:

Submission Date:	April 29, 2022
Editorial Decision:	June 6, 2022
Revision Received:	June 6, 2022
Accepted:	June 8, 2022

Editor: Alison Sinclair

Reviewer(s): The reviewers have opted to remain anonymous.

Transaction Report:

DOI: <https://doi.org/10.1128/spectrum.01447-22>

Response to Reviewers' Comments

Journal: *mBio*

Manuscript ID: mBio00405-22

Title: Plasma Virome Reveals Blooms and Transmission of Anellovirus in Intravenous Drug Users with HIV-1, HCV and/or HBV infections

Authors: Yanpeng Li, Le Cao, Mei Ye, Rong Xu, Xin Chen, Yingying Ma, Ren-Rong Tian, Feng-Liang

Liu, Peng Zhang, Yi-Qun Kuang, Yong-Tang Zheng, Chiyu Zhang

Response to Editor comments

It is important to address the concerns about alignment of short reads and qPCR or other validation of results.

Response: Generally, the reviewers mentioned the sensitivity of our method. We provided a new Table S1 for the NGS recovering rate of the original (RT-)qPCR results for the detection of HIV-1/HCV/HBV. NGS showed a good recovering rate (94.7%). We think the small discrepancy of these three viruses' detection would not influence our results, as the most abundant virus in blood was anellovirus. We also provided the correlations of qPCR and NGS reads for anelloviridae, HIV-1, HCV and HBV. Correlations were observed for anelloviridae and HBV, but not for HIV-1 and HCV. These data were provided in the revised ms, and several points of limitations were added in the discussion.

Besides, to achieve accurate annotation of NGS short reads, we used both assembled contigs and reads for the virus annotation (many studies only used short reads for the analysis, such as *Legoff et al. Nat Med 2017*; *Gu et al. Nat Med 2021*, and virus candidates were further checked for potential false positives. We believe our methods are both stringent and sensitive enough. The methods were also described in previous studies by us and others (*Li et al. J Virol 2021*; *Li et al. Viruses 2020*; *Li et al. AIDS 2020*; *Liu et al. mSphere 2021*; *Siqueira et al. Nat Commun 2018*; *Zhao et al. Virology 2017*).

In addition, we'd like to thank both reviewers' suggestions and comments on other points, and we've made corresponding revisions to our manuscript.

Response to Reviewer #1 Comments

Li and coworkers characterize the virome of persons who inject drugs using metagenomic approach and reveal the vast diversity, especially of the annelloviruses. The number and diversity contribute to our understanding of this relatively understudied area of human health.

Since the goal is to characterize the virome, the major concern with the work is the lack of validation of the findings. Overall, this could be done by repeating the work on another blood specimen to reproduce the same or very similar virome. The paper reports 22 PWID with circoviridae. That is an interesting finding that might be mapped to porcine viruses and confirmed in another blood specimen.

Response: Thanks for the suggestion. We agree that a confirmation with another cohort would be great, but recruiting new individuals and collecting samples are not available at this moment. In our previous study of another high-risk population (MSM) (*Liu et al. mSphere 2021*), we found similar results

that HIV infection would greatly increase the abundance and diversity of anellovirus. We added a few points about the relationship with our previous study, and also mentioned the limitations of the methods used. Line: 428-437.

Thanks for the suggestion. We also checked the results of circoviridae, a full viral genome (GenBank: ON226770) that were most closely related to porcine circovirus 3 (<71% identity) were found in one IDU sample. As high-risk populations, IDUs may carry emerging or neglected viruses. It would be interesting to screen more IDUs and/or other high-risk individuals for their prevalence, and investigate the possible origin and transmission of this virus. A few points were added in the discussion. Line 423-427.

On the interpretation of the data, the authors should not assume causality. You have no evidence that the HBV, HCV, or HIV infections caused anything and you have absolutely no data on immune responses. It seems a PWID who already acquired one or more virus infection in blood not surprisingly will have others. Did those PWID get HIV, HCV, or HBV because they were more immunosuppressed than the PWID who didn't? It is widely accepted that they acquire those infections because they are more exposed. More than likely, it will be the same story for anellovirus but, until proven, it may be wise to just observe what you found and not speculate (with no evidence) on how.

Response: Thanks for the suggestion. In our revised MS, we removed several discussions to avoid arbitrary speculations.

How did the authors deal with alignment of the short reads produced by the Novaseq. Given the lack of clear reference sequences, how did you train Megahit to correctly align your sequences? Could some of the apparent diversity be due to problem with alignment? How do you perform the phylogenetic analysis when you aren't confident you are comparing viruses with a common ancestry? For example, one wouldn't include dengue and HCV together to make transmission inferences. How did you come up with your thresholds for clusters?

Response: Megahit was described in 2015 (*Bioinformatics*. 2015 May 15;31(10):1674-6), and is the most popular used program in metagenomics. It was tested by many studies for having good performance (such as *BMC Genomics*. 2021 Nov 24;22(1):849; *Genes Genomics*. 2019 Sep;41(9):1077-1083; *Brief Bioinform*. 2020 May 21;21(3):777-790; *J Microbiol Methods*. 2018 Aug;151:99-105).

To achieve accurate annotation of NGS short reads, we used both assembled contigs and reads for the virus annotation (many studies only used short reads for the analysis, such as SURPI method, *Legoff et al. Nat Med* 2017; *Gu et al. Nat Med* 2021), and virus candidates were further checked for potential false positives. We believe our methods are both stringent and sensitive enough. The method was also described in previous studies by us and others (*Li et al. J Virol* 2021; *Li et al. Viruses* 2020; *Li et al. AIDS* 2020; *Liu et al. mSphere* 2021; *Siqueira et al. Nat Commun* 2018; *Zhao et al. Virology* 2017).

The phylogenetic analysis was performed for anellovirus and pegivirus separately, and only related virus sequences (either anellovirus or pegivirus) were included in each phylogenetic tree. In order to determine the possible transmission traits of anellovirus and pegivirus, we first analyzed the phylogenetic relationship of all pegiviruses, and found many sequences were clustered closely (similar to other bloodborne viruses, e.g. HIV and HCV), and then we identified transmission clusters with a stringent threshold of 1% genetic distance (generally 1.5-4.5% distance was used for HIV-1 and HCV; *J Infect Dis*. 2014 Jan 15;209(2):304-13.; *Sci Rep*. 2016 Oct 4;6:34729; *Hepatology*. 2021 Oct;74(4):1782-94.). Pegivirus and HCV relate to each other on the phylogenetic tree and share similar genomic structures, so the threshold of 1% distance is stringent and accurate for cluster picking.

After identifying transmission clusters based on the pegivirus phylogeny, we compared the diversity and genetic distance of anellovirus among individuals within each cluster as well as between clusters. So, the transmission of anellovirus was not determined by direct comparison between pegivirus and anellovirus, instead it was determined through the analyzing of the relatedness of the anellovirus between linked individuals (also see recent study: *Abbas et al. Am J Transplant. 2019 April; Kandathil AJ, et al. Nat Commun. 2021.*)

We added more descriptions in the method of the revised ms to clarify this. Line: 164-170.

You appear to have detected reads for the the already recognized viruses. To what extent did the abundance in the HBV group relate to simply HBV DNA reads vs some intrinsic difference in the group? You might tell us exactly what you detected to give us a sense of your overall sensitivity. For example, did you correlate qPCR for HBV, HCV, and HIV with your metagenomic read number?

Response: Metagenomic sequencing is widely used for scientific purpose, and more and more in clinical practice. Many studies have evaluated NGS as a powerful tool as compared to those traditional methods. We compared the detection (positive or negative) of these viruses by both methods, and NGS could reach a generally high consistency with the qPCR results: 24(NGS)/26(PCR) for HIV+ group, 11/11 for HCV+ group, 28/29 for both HIV+ and HCV+ group, 9/10 for HBV+ group. These data indicated that we had a good sensitivity.

We also agree that it is helpful to correlate the viral reads with qPCR viral loads to validate the NGS results. Unfortunately, the initial PCR screening was not done by quantitative method, so we used the remaining samples (several samples had run out) to measure TTV, as well as HIV, HCV, and HBV again with qPCR methods. We found significant correlation for TTV Ct value and reads. However, for the other three viruses, significant correlation was only observed for HBV. Because we used a WTA kit for the enrichment of viral nucleic acids (common in virome studies), it may preferentially amplify TTV genomes (circular) and disturbs the correlations of HIV and HCV. We added a few points of the limitations of the methods used. Line: 423-427.

A new Table S1 and Figure S1 of the comparison between NGS and qPCR was provided in the revised ms.

New Table S1:

PCR positive	NGS positive	Recovering rate by NGS
HIV-1 (n=26)	24	92.3%
HCV (n=11)	11	100%
HIV-1&HCV (n=29)	28	96.5%
HBV (n=10)	9	90%

New Figure S1. Correlations between qPCR and NGS reads:

The paper needs some grammatical assistance in English, which is understandable (and available).

Response: Thanks for the suggestion. We carefully checked and revised our languages throughout the manuscript.

Response to Reviewer #2 Comments

Li et al examines the human virome of intravenous drug users using metagenomic sequencing and describes how the virome differs between healthy individuals and patients with chronic viral infections (HIV, HBV, HCV). The authors report higher viral burden and diversity in IDUs and further viral abundance in IDUs infected with HIV, HCV, and HBV. They describe blooms of anelloviruses in IDUs compared to controls, confirming the interesting and underexplored role of these ubiquitous viruses in humans.

Some questions/suggestions that I think will strengthen this manuscript.

1. Many of the results describe the viral composition aggregated across all study participants or groups of study participants. Please present the presence of virus reads broken down by individual.

Response: Thanks for the suggestion. We provided a new Figure S2 (related to Figure 1b&c) that displayed the relative abundance of virus reads by each individual.

New Figure S2. Relative abundance of main vertebrate viruses (a) and prokaryotic viruses (bacteriophages)(b) by individual.

2. This study examined IDUs and IDUs positive for HIV, HBV, and HCV. What viral sequencing results (RPM, assemblies) can you report from HIV, HBV, and HCV in the study participants?

Response: We compared the detection (positive or negative) of these viruses by both methods, and NGS could reach a generally high consistency with the qPCR results: 24(NGS)/26(PCR) for HIV+ group, 11/11 for HCV+ group, 28/29 for both HIV+ and HCV+ group, 9/10 for HBV+ group. These data

indicated that NGS had a good sensitivity (94.7%). Besides, we also performed the correlation analyses between qPCR and NGS reads for anelloviridae, HIV-1, HCV and HBV. We found significant correlation for TTV Ct value and NGS reads. However, for the other three viruses, significant correlation was only observed for HBV. Because we used a WTA kit for the enrichment of viral nucleic acids (common in virome studies), it may preferentially amplify TTV genomes (circular) and disturbs the correlations of HIV and HCV. We added a few points of the limitations of the methods used. Line: 423-427. (Please also see the response to reviewer 1)

A new Table S1 and Figure S1 of the comparison between NGS and PCR was provided in the revised MS.

New Table S1:

PCR positive	NGS positive	Recovering rate by NGS
HIV-1 (n=26)	24	92.3%
HCV (n=11)	11	100%
HIV-1&HCV (n=29)	28	96.5%
HBV (n=10)	9	90%

New Figure S1. Correlations between qPCR and NGS reads:

3. Please describe any steps taken in your sequencing analysis pipeline to remove false-positive reads derived from contaminants such as plasmids and vectors.

Response: Human- and bacterium-depleted sequences were first searched against viral nucleic acid and protein database, then all the viral hit candidates were searched against the NCBI nonredundant nt and nr database to remove reads or contigs that have higher similarity to sequences related to host, bacteria, fungi, plasmids, vectors and other non-viral sequences than to viral sequences (false positives). We revised the description of this point in method, line 138-141.

4. Longitudinal samples were unavailable for this study, and patient age may influence virome composition and diversity. Were any changes in the virome make-up observed by duration of drug use or patient age?

Response: Thanks for the important suggestion. We analyzed the associations of viral abundance and diversity with age and duration of drug use. We found that age didn't influence the virome composition; however, a longer time of drug use was associated with higher viral reads and richness. The data indicate that longer drug use may lead to higher risk of viral infections. A new Figure S3 was provided in the revised ms.

Figure S3. Influence of age (top) and duration of drug use (bottom) on the blood viral composition. Spearman's correlation was analyzed between age/duration of drug use and viral reads (RPM), Richness, Shannon index.

5. Previous blood virome studies report *Herpesviridae* as a common component, yet they appear largely undetected in this study (Fig1). How do the authors reconcile those findings with prior studies?

Response: *Herpesviridae* could be prevalent in general population, but most infections are asymptomatic or latent infections, which can be activated under certain disease status or impaired immune system. Patients on tissue transplantation are usually under immune suppression, and virome studies of these individuals had higher detection rate of *Herpesviridae*. See:

Temporal dynamics of the lung and plasma viromes in lung transplant recipients. PLoS One. 2018; 13(7): e0200428. (13/15);

Unmasking viral sequences by metagenomic next-generation sequencing in adult human blood samples during steroid-refractory/dependent graft-versushost disease. Microbiome. 2021 Jan 24;9(1):28. (5/25)

However, *Herpesviridae* was not detected in other virome study of tissue transplantation patients. See:

Clinical relevance of plasma virome dynamics in liver transplant recipients. EBioMedicine. 2020

Oct;60:103009. (no herpesviridae reported)

In our previous study about plasma virome of MSM individuals, we found the presence of *Herpesviridae* only with a very low fraction of all viral hits (0.00056%). See: *HIV-1 Infection Alters the Viral Composition of Plasma in Men Who Have Sex with Men. mSphere. 2021 May 5;6(3):e00081-21.*

We searched other blood virome studies of different cohorts, and most studies also found low proportions or the absence of *Herpesviridae*. For example:

Exploring the Diversity of the Human Blood Virome. Viruses. 2021 Nov; 13(11): 2322. (0.0028%)

Deep viral blood metagenomics reveals extensive anellovirus diversity in healthy humans. Sci Rep. 2021; 11: 6921. (0.017%)

Virome comparison of deferred blood donations obtained from different geographic regions in the Sao Paulo State, Brazil. *Transfus Apher Sci. 2021 Jun;60(3):103106. (no herpesviridae reported)*

Plasma virome of 781 Brazilians with unexplained symptoms of arbovirus infection include a novel parvovirus and densovirus. PLoS One. 2020 Mar 5;15(3):e0229993. (no herpesviridae reported)

AIDS alters the commensal plasma virome. J Virol. 2013 Oct;87(19):10912-5. (no herpesviridae reported)
The plasma virome of febrile adult Kenyans shows frequent parvovirus B19 infections and a novel arbovirus (Kadipiro virus). J Gen Virol. 2016 Dec;97(12):3359-3367. (3/51 positive)

Thus, the prevalence of *Herpesviridae* in many virome studies varied, and the absence of this virus could be due to low viral loads, as well as relatively low prevalence in different populations.

Minor suggestions/edits:

1. Please state more clearly in the text of the manuscript the total number of study participants.

Response: This was mentioned in the revised manuscript, line 114: “99 drug users and 11 healthy controls”.

2. Many virome studies focus on either RNA or DNA viruses, but the library prep technique employed here is intended to capture both. Please further highlight this and discuss benefits and limitations.

Response: Thanks for the suggestion. We added a few points about the benefits and limitations of the method used in this study. Line: 429-434.

3. Anellovirus ORF1 sequences can begin with non-AUG codons. Was this considered when utilizing NCBI's ORF finder tool?

Response: Yes, ORF1 sequences can begin with non-AUG codons, and some assembled ORF1 regions were not full length CDS. So, we used the “any sense codon” option for the analysis. This description was added in line 148.

4. The authors report 4487 anellovirus ORF1 sequences in their study. Please specify if these have been de-duplicated and/or clustered at an identify threshold.

Response: ORF1 sequences of anellovirus were de-duplicated with a threshold of 99% identity. This was mentioned in our revised ms. Line 149-150.

5. Figure 1a - Please move the key that describes the colors in circles A, B, and C to the top left of the figure panel.

Response: Thanks for the suggestion, the layout of Figure 1a was adjusted.

June 6, 2022

Prof. Chiyu Zhang
Shanghai Public Health Clinical Center, Fudan University
2901 Cao Lang Road, Jinshan District, Shanghai, China
Shanghai
China

Re: Spectrum01447-22 (Plasma Virome Reveals Blooms and Transmission of Anellovirus in Intravenous Drug Users with HIV-1, HCV and/or HBV infections)

Dear Prof. Chiyu Zhang:

Thank you for submitting your manuscript to Microbiology Spectrum.

After reviewing the original reviews and the new submission, although most concerns have been addressed, one remains. I consider the point the reviewers made to be important and I recommend modification before it could be accepted for publication.

The point is made several times that the increased viral burden is a result of HIV/HCV/HBV infection. It was recommended that this is changed to show that this is speculation ie increased viral burden is associated with HIV/HCV/HBV infection rather than to imply that HIV/HCV/HBV infection causes increased viral burden.

I recommend changing this in the following places:

abstract line 36 and 48/49
line 75
line 346
line 382
line 443

Link Not Available

Sincerely,

Alison Sinclair

Journals Department
Reviewer comments:

Staff Comments:

Preparing Revision Guidelines

Please return the manuscript within 60 days; if you cannot complete the modification within this time period, please contact me. If you do not wish to modify the manuscript and prefer to submit it to another journal, please notify me of your decision immediately so that the manuscript may be formally withdrawn from consideration by Microbiology Spectrum.

Response to Reviewers' Comments

Journal: *Microbiology Spectrum*

Manuscript ID: Spectrum01447-22

Title: Plasma Virome Reveals Blooms and Transmission of Anellovirus in Intravenous Drug Users with HIV-1, HCV and/or HBV infections

Authors: Yanpeng Li, Le Cao, Mei Ye, Rong Xu, Xin Chen, Yingying Ma, Ren-Rong Tian, Feng-Liang

Liu, Peng Zhang, Yi-Qun Kuang, Yong-Tang Zheng, Chiyu Zhang

Response to Editor comments

Thank you for submitting your manuscript to Microbiology Spectrum.

After reviewing the original reviews and the new submission, although most concerns have been addressed, one remains. I consider the point the reviewers made to be important and I recommend modification before it could be accepted for publication.

The point is made several times that the increased viral burden is a result of HIV/HCV/HBV infection. It was recommended that this is changed to show that this is speculation ie increased viral burden is associated with HIV/HCV/HBV infection rather than to imply that HIV/HCV/HBV infection causes increased viral burden.

I recommend changing this in the following places:

abstract line 36 and 48/49

line 75

line 346

line 382

line 443

Response:

Thanks for your suggestions. As this study is a cross sectional study, we agree it would be better to say that the increased viral loads (including anelloviruses) are associated with HIV/HCV/HBV infections. Future studies on longitudinal samples before and after HIV/HCV/HBV infections may better address this point.

We revised our manuscript where these descriptions appear:

Abstract line: 36, 47-48

Introduction line: 74-75

Results line: 228, 251-254

Discussion and Conclusion line: 345-347, 372, 380-382, 443

June 8, 2022

Prof. Chiyu Zhang
Shanghai Public Health Clinical Center, Fudan University
2901 Cao Lang Road, Jinshan District, Shanghai, China
Shanghai
China

Re: Spectrum01447-22R1 (Plasma Virome Reveals Blooms and Transmission of Anellovirus in Intravenous Drug Users with HIV-1, HCV and/or HBV infections)

Dear Prof. Chiyu Zhang:

Your manuscript has been accepted, and I am forwarding it to the ASM Journals Department for publication. You will be notified when your proofs are ready to be viewed.

Sincerely,

Alison Sinclair
Editor, Microbiology Spectrum
